# Bone Tissue Engineering through 3D Bioprinting of Bioceramic Scaffolds: A Review and Update

**DOI:** 10.3390/life12060903

**Published:** 2022-06-16

**Authors:** Ahmad Taha Khalaf, Yuanyuan Wei, Jun Wan, Jiang Zhu, Yu Peng, Samiah Yasmin Abdul Kadir, Jamaludin Zainol, Zahraa Oglah, Lijia Cheng, Zheng Shi

**Affiliations:** 1Clinical Genetics Laboratory, Clinical Medical College & Affiliated Hospital & Basic Medical College of Chengdu University, Chengdu 610106, China; ahmadtaha11@yahoo.com (A.T.K.); weiyuanyuan@cdu.edu.cn (Y.W.); wanjun0726@163.com (J.W.); 13378119012@163.com (J.Z.); pengyu1216@163.com (Y.P.); 2Faculty of Medicine, Widad University College, Kuantan 25200, Pahang, Malaysia; samiah.abdulkadir@tdmberhad.com.my (S.Y.A.K.); drjamaludinzainol@gmail.com (J.Z.); 3School of Science, Auckland University of Technology (AUT), 55 Wellesley Street East, Auckland 1010, New Zealand; phk9904@autuni.ac.nz

**Keywords:** 3D bioprinting, bone tissue, bioceramic, scaffold, tissue engineering, regenerative medicine

## Abstract

Trauma and bone loss from infections, tumors, and congenital diseases make bone repair and regeneration the greatest challenges in orthopedic, craniofacial, and plastic surgeries. The shortage of donors, intrinsic limitations, and complications in transplantation have led to more focus and interest in regenerative medicine. Structures that closely mimic bone tissue can be produced by this unique technology. The steady development of three-dimensional (3D)-printed bone tissue engineering scaffold therapy has played an important role in achieving the desired goal. Bioceramic scaffolds are widely studied and appear to be the most promising solution. In addition, 3D printing technology can simulate mechanical and biological surface properties and print with high precision complex internal and external structures to match their functional properties. Inkjet, extrusion, and light-based 3D printing are among the rapidly advancing bone bioprinting technologies. Furthermore, stem cell therapy has recently shown an important role in this field, although large tissue defects are difficult to fill by injection alone. The combination of 3D-printed bone tissue engineering scaffolds with stem cells has shown very promising results. Therefore, biocompatible artificial tissue engineering with living cells is the key element required for clinical applications where there is a high demand for bone defect repair. Furthermore, the emergence of various advanced manufacturing technologies has made the form of biomaterials and their functions, composition, and structure more diversified, and manifold. The importance of this article lies in that it aims to briefly review the main principles and characteristics of the currently available methods in orthopedic bioprinting technology to prepare bioceramic scaffolds, and finally discuss the challenges and prospects for applications in this promising and vital field.

## 1. Introduction

Bone defects or injuries from trauma, infection, tumors, and congenital diseases cause patients to lose their ability to do basic exercises and greatly affect their quality of life. Healthy bones are essential for vital functions of the human body. Although bone has an excellent intrinsic repair capacity, its ability to fill very large defects remains complex and limited. Bone tissue is composed of two different structures: cancellous and cortical bone [1,2,3]. The internal structure of cancellous bone is spongy and has a porosity of 50% to 90%. Cortical bone is a dense outer layer of bone with a porosity of less than 10%. Both types of bone formation need to undergo a process of dynamic remodeling, maturation, differentiation, and resorption. These processes are controlled and regulated by the interaction between bone cells: osteoblasts and osteoclasts. Osteoblasts are mainly responsible for the formation of new bone, while osteoclasts are responsible for the resorption of old bone. This dynamic process involving osteoclasts and osteoblasts is called ‘bone remodeling’, and it is of great significance for promoting bone regeneration and maintaining the integrity of tissue structure [4,5]. These patients need to surgically transplant bone substitutes to the bone defect site to help bone stabilization and regeneration. In Europe, the growth rate of fractures is expected to reach 28% from 2010 to 2025 [6]. The number of age-related fractures in the United States is expected to increase from 2.1 million in 2005 to more than 3 million in 2025 [7]. Bone is the second most frequently transplanted tissue in the world, and bone grafts and bone substitute materials are used in at least 4 million operations every year [8,9,10].

Therefore, bone repair and regeneration are difficult challenges orthopedic, craniofacial, and plastic surgeons face in instances of bone loss. Methods for repairing bone defects include either bone tissue transplantation or the use of synthetic materials and natural derivatives. The source of the transplanted bone can be divided into three categories: (1) Autologous bone transplantation, which is the internationally recognized ‘gold standard’ of bone implants, which has the advantages of good histocompatibility and no disease transmission, but there are secondary operations and infections possibilities at the donor site. It is usually associated with problems such as limited bone supply, long operation time, and severe limitations in bone shape. (2) Xenograft; this type of bone grafting is derived from animals. The supply is relatively abundant, the endogenous antigenic substance has been inactivated, but there is still the risk of immune rejection. It is subject to religious and moral rejection and is not widely developed. (3) Allogeneic bone transplant is sourced from other people’s donations and the supply is relatively wide, but there is a potential risk of spreading infectious diseases such as HIV and hepatitis B. Furthermore, one of the most important complications of this type is transplant rejection. Moreover, bone grafts such as artificial hip joints have a limited functional life and cannot promote bone tissue regeneration [10,11,12,13]. Therefore, there is a growing focus on the field of regenerative medicine to provide new and alternative methods of bone grafting to solve these problems.

However, designing biomaterials that meet the different needs of bone tissue engineering materials in line with their different functions is a major challenge. It requires scientists to have a sufficient and deep understanding of the components of normal bone tissue. The structure and composition of normal bone tissue are very complex; bone tissue is made up of natural materials such as type 1 collagen and hydroxyapatite. This mineralized, viscous, and elastic connective tissue performs a vital function in supporting and protecting other tissues in our body. Structurally, bone tissue is characterized by a complex and highly organized hierarchical structure (Figure 1). These structures are highly compatible with bone functions throughout the body [10,11,12,13,14,15].

Scientists have explored large-scale materials and different types of natural and artificial biomaterials to maximize bone tissue regeneration. Biomaterials play a pivotal role in this process, serving as an essential scaffold for the growth of new bone. These materials which are used as bone-like biomaterials include main materials derived from metals, polymers, ceramics, and natural materials. Polymers are divided into natural polymers and synthetic polymers, and natural polymers belong to “natural materials” [16,17,18,19]. There must be many things in biological materials to be safely incorporated into the human body for a long time, such as non-toxicity and biocompatibility, as well as appropriate physical and chemical stability. Artificial synthetic materials or natural bio-derived materials have the advantages of abundant supply sources, no antigenicity or little antigenicity, and the ability to be manufactured according to the needs of the bone defect repair. However, there are disadvantages such as the inability to bear weight in the early stage, and it is easy for the implant site to collapse, which restricts its application scope and curative effect to a certain extent. On the other hand, traditional medical metal materials lack the necessary biodegradability and biological activity, and their elastic modulus is much higher than that of human bone, which is prone to stress shielding effect, causing risks such as bone resorption, and atrophy around the implant [16,17,18,19].

The development of bone tissue engineering provides new ideas for bone defect repair: making bone repair scaffolds a new research focus. The bone repair scaffold is the core component of bone tissue engineering and plays a very important role in bone defect repair. Bone repair scaffolds can provide a suitable adhesion and proliferation environment for cells. With the degradation of scaffold biomaterials, the deposition of the extracellular matrix can achieve the purpose of repairing bone defects [20,21,22,23]. Therefore, an ideal bone repair scaffold must have the following characteristics: (1) Good biocompatibility, which is conducive to cell adhesion, proliferation, and differentiation; (2) Degradability, and the degradation rate can match the regeneration and repair speed of human bone tissue; (3) The internal three-dimensional, interconnected pore structure can not only meet the exchange of nutrients and metabolites but also guide cell growth; and (4) It has a certain strength and can meet the early mechanical requirements of the scaffold implantation site [20,21,22,23]. Accordingly, the continuous development of a biomaterial scaffold is one of the requirements to reach the optimal material and an efficient method for biomaterials in bone tissue engineering. Recent advances have continued steadily in the types and scope of the application of 3D printing materials for orthopedic bioceramic scaffolding. Different methods have their advantages and disadvantages, but one common factor is one goal that seeks to print effective bone tissue that holds hope for many to ease their suffering [20,21,22,23].

This article briefly reviews the principles and characteristics of 3D printing technology for preparing porous scaffolds and focuses on active component loading, micronutrient doping, functional surface modification, porous structure optimization, scaffold surface micro/nanostructure, and biomimetic multilayer structure construction, and other strategies that have improved the function of 3D-printed bioceramic scaffolds. This review also explores current research challenges and looks at the application prospects of bioceramic 3D printing. Figure 2 shows a schematic illustration of the application in 3D bioprinting.

## 2. Bioceramic 3D Printing Overview

### 2.1. Biomaterials

The currently used bone tissue engineering scaffold biomaterials can be divided into three categories: natural polymer materials, artificial synthetic materials, and composite materials. Natural polymer materials include chitosan, alginate, and collagen, which have the advantages of good biocompatibility, no immunogenicity, and non-toxic degradation products (Table 1). However, the mechanical properties of natural high molecular polymers are poor, and the degradation rate is too fast, so they are rarely used in the preparation of a single scaffold [24,25,26]. Artificial synthetic materials include polylactic acid, polylactic acid, and polycaprolactone (PCL). These are biologically inert materials with weak hydrophilicity and cell adhesion ability. Among them is PCL, a widely used synthetic polymer polyester material with good biocompatibility, degradability, and certain mechanical strength. It is approved by the U.S. Food and Drug Administration (FDA), a class of biopolymer materials that can be used clinically.

In recent years, PCL has been widely used in the preparation of bone repair scaffolds. However, the surface of pure PCL material lacks cell affinity sites, the biological activity is insufficient, and it is difficult to meet the needs of bone implantation [27,28]. In addition, PCL material has a slow degradation rate and a long degradation cycle. It takes more than 2 years to completely degrade, which is difficult to match the repair speed of human bone tissue and is not conducive to later healing [29,30,31]. Therefore, because a single material cannot meet the requirements of an ideal scaffold material, the preparation of composite materials has become a new research focus of bone scaffolds. Composite biological materials can have the advantages of multiple materials at the same time. While improving the biological properties of the materials, they can also significantly promote bone regeneration [32,33,34,35]. In addition to good biological activity, an ideal bone repair scaffold should also have an internally connected porous structure. Porous scaffolds can achieve the adhesion of certain specific proteins, including vitreous binding protein and fibronectin, and provide a suitable environment for cell adhesion and internal growth on the surface of the material, thereby promoting the repair of bone defects. Therefore, there are many methods for preparing three-dimensional porous bone tissue scaffolds, such as the phase separation method, solution-particle leaching method, gas foaming method, and wire mesh forming method [36,37,38,39,40,41,42]. These methods all need to be prepared manually, and the internal pore structure of the prepared scaffold is unstable, with poor reproducibility, and it is difficult to have a personalized geometric shape. In addition, the traditional preparation process is relatively complicated and is not suitable for large-scale clinical promotion. Therefore, porous PCL bone tissue engineering scaffolds prepared by traditional methods cannot meet clinical requirements in porosity, elastic module, and defect matching rate, which severely limits the popularization and application of such bone tissue engineering scaffolds [36,37,38,39,40,41,42].

### 2.2. Bioceramic Scaffolds

Ceramic, on the other hand, is considered a serious challenge for most 3D printing techniques, as its melting points are very high, and its optical and mechanical properties are not considered ideal for most of the current printing technologies. One of the most popular methods of printing ceramics today is the introduction of ceramic powders into the monomer liquids that polymerize and harden with light. Thus, the ceramic powder between plastics is in amounts typically up to 50 percent of the total weight of the printed material [43,44,45,46]. This process, as it is now, does not give the printed material attractive ceramic properties such as hardness and strength, but rather the properties of the plastic that sticks the ceramic powder atoms together. To obtain high-purity ceramics, the printed object is heated to a high temperature based on the analysis of plastic and the resulting carbon oxidation and volatilization in the form of carbon dioxide gas, thus reaching high purity ceramics that shrink in shape as a result of the decomposition of plastic (Figure 3). With medical advances in the use of biomaterials for orthotics, various industrial applications have moved to the development of materials with the ability to interact with the biological environment and elicit specific biological responses. Bioceramic materials have excellent biocompatibility and biological activity, with stable physical and chemical properties [43,44,45,46]. They are also low cost and easy to mass-produce, and so are widely used in the field of regenerative medicine. Bioceramics are materials such as alumina, zirconia, bioactive glass, glass-ceramics, hydroxyapatite, resorbable calcium phosphates, and others (Table 2).

These materials have good biological activity and can promote the differentiation of bone marrow mesenchymal stem cells into osteoblasts in vitro and induce new bone formation in vivo. Bioceramic materials can effectively provide mechanical support [52,53,54,55]. However, simple bioceramic powder cannot be used directly to repair bone defects of a certain size due to rapid deterioration and easy loss. It can only then be prepared in a porous 3D tissue engineering scaffold, promoting nutrient exchange, and inducing the growth of new bone tissue. It can then be used to repair and regenerate large-scale bone defects. These include hydroxyapatite, calcium phosphate, calcium sulfate, and bioactive glasses and are the most common and clinically available bone substitute materials. Hence, bioceramic scaffolds represent a cornerstone for bone regeneration, as they have distinct properties that emerge as promising alternatives to bone grafts [62,63,64,65].

Scaffolding provides mechanical support until new tissue formation is completed in the affected area. They are 3D porous matrices that act as temporary templates for cell adhesion and proliferation. In the preparation of tissue engineering scaffolds, traditional preparation techniques—such as the pore-forming agent method, organic foam template method, freeze-drying method, and foaming method—can prepare porous bioceramic scaffolds with high porosity. However, these traditional techniques are difficult to produce a three-dimensional scaffold that matches the defect tissue and has a complex macroscopic structure, and it is even more difficult to accurately control the geometric shape and size of the micron-scale porous structure inside the scaffold. In addition, the pore structure obtained by these technologies is often heterogeneous and does not have a uniform shape and size. Compared with traditional processes, 3D printing technology can quickly and efficiently print complex shapes with customized three-dimensional scaffolds, and it can also precisely control the porous structure of the scaffold on the micrometer scale. The current 3D-printed bioceramic scaffolds have a single function: low osteogenic potential, and difficulty in inducing angiogenesis [11,12,13,14,15]. These problems severely limit the application of 3D-printed bioceramic scaffold materials in the biomedical field. To improve the biological performance of 3D-printed bioceramic scaffolds and give them more biological functions, materials scientists have taken several strategies from element doping, surface modification, and hierarchical and bionic structure construction.

### 2.3. 3D Printing Manufacturing Technologies

In the case of printing biomaterials, there are usually two ways to form the organic material: either direct or indirect printing. Direct printing is done by spreading certain cells (stem cells, for example) in a gelatinous liquid suitable for the cells to live and grow. The material and cells they contain are drawn by printing to form the required model, and then these cells are provided with nutrients and an appropriate environment to keep them alive. The indirect method relies on the principle of printing a scaffold in the shape of the desired organ and spreading the cells around the shape after printing so that the cells climb and grow and take the printed shape. In recent years, the additive manufacturing (AM) technology represented by 3D printing technology has received extensive attention and research [66,67,68,69,70]. It involves many methods, some of which are deposited in layers, while others work by hardening polymers. This is in conjunction with the slide-slicing software where material selection allows control of pore size, porosity, and architecture. Compared with the traditional porous material preparation process, the biggest advantage of 3D printing technology is that it can accurately control the structure of the bioceramic scaffold (including the size and shape of the internal pores, and the overall shape of the scaffold) from micro to macro scale. This feature enables 3D printing technology to design tissue engineering scaffolds to repair tissue defects for patients according to actual needs, to achieve precision medicine [9,10,11,12,13,14,15]. The most used 3D printing techniques are briefly described in Table 3.

There is generally no single bioprinting method that enables the production of all the required synthetic tissue scales and complexities, as each method has specific strengths, weaknesses, and limitations. At present, the commonly used 3D printing technologies are inkjet 3D printing technology, selective laser sintering technology, ink direct writing 3D printing technology, and stereolithography (SLA) printing technology. The following will briefly summarize the principles, advantages, and disadvantages of these 3D printing technologies. A brief comparison of these methods can be seen in Table 3.

#### 2.3.1. Inkjet 3D Printing Technology

Inkjet printing technology is also known as 3DP technology. Its principle is to spray adhesive on specific areas through the print head to bond the powder materials together and then accumulate them layer by layer leading to the final scaffold embryo. When the printer is running, the print head can accurately move in the three dimensions of X, Y, and Z according to the instructions generated by the computer-aided design (CAD) file and spray the adhesive in the designated area. After printing a layer, the printing platform will move down, and at the same time, the new powder will be spread evenly on the previously printed support by the roller. This cycle repeats until the entire bracket is printed. After the printing is completed, the operator removes the unbonded powder and can obtain a three-dimensional structure of the scaffold body for subsequent sintering. In the experiment, the size, morphology, surface roughness, and wettability of the powder, as well as the concentration, viscosity, and volume of the binder droplets will affect the quality of the final printed scaffold embryo [10,11,12].

The main advantages of inkjet 3D printing technology are the low cost, wide application range of materials, and no additional support required for printing. However, the disadvantages of inkjet 3D printing technology are also obvious. For example, the mechanical properties of the printed bracket are relatively low, the surface of the bracket is very rough, and the unbonded powder may be trapped inside the scaffold. Moreover, the printing accuracy is poor compared with the direct-ink-writing (DIW) and SLA 3D bioprinting technologies [10,11,12].

A very recent development is omnidirectional ceramic bioprinting in cellular suspensions. In research conducted at the University of New South Wales in Australia, a scientific team was able to 3D print bone parts with living cells with multidirectional ceramic bioprinting, enabling them to repair damaged bone tissue that hardens within minutes when placed in water. It is a pioneering experiment and the first time that such materials can be created at room temperature with living cells and without harsh chemicals or radiation [71]. This study demonstrates that within a high density of stem cells, directed by cellular regulation, osteogenesis could be promoted in vitro. Where, through the local nanocrystallization mechanism of its components in aqueous environments, the inorganic ink is converted into mechanically interlocked orthopedic nanocrystals. This technology has the potential to radically update current practices to alleviate the suffering of patients who lose part of the bone tissue, as the anatomical structure of their bones is matched and printed directly in the cavity with their cells. Rapid advances in this field are leading to the development of new applications and materials in printing technologies that enable speed and accuracy for multi-material printing [72,73,74,75].

#### 2.3.2. Selective Laser Sintering Technology

This technique is known as selective laser sintering (SLS) as it is based on that principle and is usually associated with nylon powders or what is combined with nylon from carbon or sometimes glass and others. Like inkjet 3D printing technology, SLS technology is also a powder-based printing technology. The difference is that the inkjet 3D printing technology uses a liquid adhesive to bond the powder and then print the support layer by layer. The SLS technology uses a laser to heat the polymer coating on the powder surface or directly fuses the powder to print a complete scaffold layer by layer. The technology of printing the scaffold by directly melting the powder is called the direct SLS technology; the technology of printing the scaffold by melting the polymer coating on the surface of the powder is called the indirect SLS technology [76,77]. During the printing process, in addition to the size and shape of the powder and the type and amount of polymer coating that will affect the printing results, the laser used by the printer is also the most important factor. The power of the laser, size of the spot, irradiation time, and scattering and heat conduction caused by the powder will all affect the printing effect of the final scaffold [78].

However, SLS technology has many insurmountable limitations, such as the high cost caused by using lasers, the rough surface and low resolution of the scaffold caused by the thermal diffusion of the powder after the laser is irradiated, and the long printing time [78]. Although SLS results in relatively weak structures, this technique has promising prospects by focusing current research on how to improve robustness without sacrificing accuracy. Another technology, although similar to SLS, is selective laser melting (SLM) which is based on the method of melting powder using a high-powered laser applied to manufacture exceedingly customized, value-added parts with specific surface morphological features [79]. However, Ponnusamy et al. have found that the same material, when printed using different machines, produced different mechanical and microscopic properties for many reasons. Therefore, the repetition of the mechanical performance of printed parts using different machines requires further research and comparison [80].

#### 2.3.3. Direct-Ink-Writing 3D Printing Technology

Direct-ink-writing (DIW) technology is also known as ‘robocasting’. DIW technology is different from inkjet 3D printing technology and SLS technology based on powder materials. The DIW technology directly extrudes a water-based colloidal suspension (printing ink, also known as printing paste) layer by layer by moving the print head to construct a three-dimensional scaffold. The printing paste used in DIW technology needs to have the characteristics of shear-thinning, and it needs to maintain a three-dimensional shape after being extruded without collapsing [81]. The printing effect of a pneumatic DIW printer is affected by parameters such as air pressure, print head moving speed, and the distance between the nozzle and the printing table. In addition, the solid content of the printing slurry, slurry viscosity, ceramic particle size, and morphology will all affect the final printing effect [82].

The most prominent advantages of DIW technology are fast printing speed, easy operation, low cost, and good printing accuracy and it can be widely applied to various material systems. These characteristics make DIW technology widely used in the preparation of three-dimensional porous scaffolds [83]. However, DIW technology also has some disadvantages that cannot be ignored. For example, the structural elements of the scaffold printed by DIW technology is a cylindrical pillar with a certain diameter, which makes the printing accuracy of DIW technology lower than that of SLA printing technology. For some complex structures, DIW technology requires additional support to assist in printing; during the printing process, the support may be dented and deformed [9,10,11,12]. These disadvantages limit the application of DIW technology, making it unsuitable for the preparation of higher precision materials.

#### 2.3.4. SLA Printing Technology

Different from other 3D printing technologies, the SLA printing technology prints the 3D scaffold layer by layer by light-induced polymerization of photosensitive resin. The principle of photopolymerization enables SLA technology to have extremely high printing accuracy compared to other technologies. SLA technology is mainly used for printing polymer materials. However, researchers can add the ceramic powder to the slurry system to print a ceramic–organic composite scaffold, and then remove the organic matter through sintering treatment to obtain a pure ceramic phase 3D scaffold [84]. The parameters that affect the printing effect of SLA technology include the resolution of the light source, exposure time, optical power, type of resin in the printing paste, size and morphology of the ceramic powder, solid content, and type and amount of dispersant [85]. The control precision and accuracy of SLA technology on the shape and size of the inner whole structure of the scaffold is unattainable by other printing technologies. However, SLA technology also has some unique disadvantages. For example, when printing some complex structures, additional support is required. The removal of these supports consumes a lot of time and energy and affects the surface roughness of the support; after printing, there are residues in the support. For uncured slurries, cleaning these slurries containing toxic resin monomers is very troublesome and time-consuming [85]. Bio ink crosslinking mechanisms and application strategies in extrusion-based 3D bioprinting are shown in Figure 4.

#### 2.3.5. Fused Deposition Modeling (FDM)

It is among the frequently used types of 3D printing techniques and stands out from the rest as the most common working principle for incorporating natural fiber reinforced polymer composites. FDM uses thermoplastic compounds that allow layers of parts to be fabricated to produce complex shapes and geometries using enhanced mechanical properties [86]. One of its advantages is that the composite filaments facilitate 3D printing without or with the least possible change in the parts of the devices. Moreover, the use of polymer composite filaments reinforced with natural fibers has drawn attention to this method as it is environmentally friendly, highly degradable, and economical. However, printing using pure polymer leads to the low performance of this method, which is one of the reasons that limit the scope of its applications as well. Other limitations include uneven printing, heterogeneous distribution of the interface of the fiber matrix, and weak connection between the layers [87].

## 3. Improvements in 3D Printing Technology for Preparing Bioceramic Scaffolds

The rapid development of additive manufacturing technology represented by 3D printing technology has made it possible to customize clinical solutions for bioceramic scaffolds according to the actual needs of patients. Various advanced 3D printing technologies can design and prepare three-dimensional porous scaffolds that are completely consistent with the tissue defect based on clinical data, and greatly reduce the time and cost of product design, processing, and application. However, traditional 3D-printed bioceramic scaffolds still have many disadvantages, such as single function, low bone formation efficiency, and difficulty in inducing ingrowth of new blood vessels. These problems make 3D printing materials for bioceramic scaffolds not a typical alternative to current clinical solutions, but rather they need some time to become so. To improve the biological performance of 3D-printed bioceramic scaffolds and construct multifunctional scaffolds to broaden their applications in the biomedical field, scientists systematically studied the loading activity from the perspectives of the composition and structure of the material. This includes composition, doping with trace elements, surface functional modification, optimization of porous structure, construction of a micro-nano-meter structure, and many other strategies. With the deepening of research, high-performance, multi-functional 3D-printed bioceramic scaffolds tailored for patients will eventually become a reality.

### 3.1. Improvements in Material Components

Bioceramic scaffolds can release biologically active ions in the body, such as calcium ions (Ca^2+^), phosphate ions (PO4^3–^), and silicate ions (SiO4^4–^). These ions can change the behavior and fate of cells by activating specific signaling pathways in cells and are conducive to the repair and regeneration of damaged tissues [88]. However, the traditional 3D printing bioceramic scaffold material has a single component, which also leads to its low bone formation efficiency and fewer functions. To improve the bone-forming ability of 3D-printed bioceramic scaffolds—endowing them with angiogenic, antibacterial, and anticancer capabilities—materials scientists mainly use active ingredients, doping with micronutrients, and modify functional coatings in optimizing the strategy. The introduction of these functional ingredients can greatly improve the biological properties of 3D-printed bioceramic scaffolds, and it is also possible to make them have antibacterial and tumor treatment functions that were not available before, making them multifunctional tissue engineering scaffolds.

#### 3.1.1. Carrying Active Ingredients

Loading on a bioceramic scaffold is an effective method to improve the biological performance of the scaffold. The 3D-printed bioceramic scaffolds loaded with drugs can not only provide climbing sites to promote the adhesion and growth of tissue cells, but also serve as a stable carrier to ensure the long-term release of drugs without rapid loss. The combination of these features makes the medicinal device formed by drugs and 3D-printed bioceramic scaffolds have a very broad prospect in the field of regenerative medicine. Bone morphogenetic protein-2 (BMP-2) can regulate the differentiation of mesenchymal stem cells into osteoblasts and ultimately induce the formation of new bone [89]. Ishak et al. [90] used DIW printing technology to prepare a 15% hydroxyapatite (HAP): 85% β-TCP biphasic scaffold and sintered it at a high temperature. Cell uptake of adenosine to increase local adenosine levels, saline, and BMP-2 were carried on different scaffolds. The authors transplanted the scaffold into a 3 mm diameter skull defect in adenosine A2A receptor knockout (A2AKO) mice and analyzed the samples at the second, fourth, and eighth weeks. The results of the study show that the scaffold carrying BMP-2 can significantly enhance the bone formation ability of the scaffold compared to other scaffolds and has an obvious repair effect in the critical size bone defect model. Similar to growth factors, some drugs have a similar ability to improve the ability of the scaffold to form bones [91]. Researchers have used an innovative low-temperature 3D printing technology to prepare a PLGA (Poly (lactic-co-glycolic acid))/TCP/icariin (PTI) composite scaffold loaded with icariin (icariin).

The addition of icariin can effectively enhance the mechanical properties of the composite scaffold and is conducive to the osteogenic differentiation of MC3T3-E1 cells and the growth into the PTI scaffold. The PTI scaffold shows better biodegradability, biocompatibility, and osteogenic ability than the PLGA/TCP (PT) scaffold without icariin. In addition, in the rabbit model of steroid-related osteonecrosis, the PTI scaffold can significantly promote the growth of new bone and the formation of blood vessels compared to the PT scaffold. In addition to carrying biological factors or drugs that promote osteogenesis, researchers can also impart antibacterial functions to the bioceramic scaffolds by carrying antibacterial drugs, thereby effectively avoiding surgical failures caused by infections that occur during surgery. Sun et al. [92] prepared a calcium phosphate ceramic scaffold loaded with berberine through ink direct-write 3D printing technology and in-situ cross-linking technology. This regulated the degree of cross-linking of the scaffold and the release profile of berberine. In vitro experiments show that the berberine supported by the calcium phosphate ceramic scaffold has obvious antibacterial effects, and at the same time exhibits low cytotoxicity and can promote the adhesion and the proliferation of MC3T3 cells. In addition to carrying a single drug, researchers can also improve the design of the 3D scaffold to load different antibacterial drugs on different parts of the scaffold, to achieve the sequential release of multiple drugs and the effect of multi-component combined sterilization. Garcia-Alvarez et al. reported a 3D-printed drug-loaded scaffold with a hierarchical structure. Wherein rifampin was loaded into the medium pores of the nanocomposite bioceramic, and levofloxacin was loaded into the printing paste in polyvinyl alcohol (gelatin-glutaraldehyde, PVA), while vancomycin was loaded into the outer gelatin layer of the scaffold [93]. The three antibacterial drugs show different release kinetics due to their different locations. The test results show that, as time goes by, the scaffold material first rapidly releases rifampicin, and then continuously and slowly releases two drugs, levofloxacin, and vancomycin. In subsequent biological experiments, this composite drug-loaded scaffold has destroyed the biofilms of gram-positive bacteria, and gram-negative bacteria and inhibited cell growth. It also showed better cell compatibility with the ability to repair defects in the bone. Figure 5 shows a schematic diagram of the deferoxamine (DFO) bridging on the surface of a 3D-printed PCL scaffold and its biological function for bone regeneration in a bone defect model.

#### 3.1.2. Doping with Trace Elements

The various trace elements, despite their small amount, have an important role in maintaining bone health and its normal structure. There is a carbon (C), hydrogen (H), oxygen (O), nitrogen (N), phosphorus (P), and other elements in the human body. Bones also contain lithium (Li), copper (Cu), strontium (Sr), manganese (Mn), iron (Fe), zinc (Zn), cobalt (Co), and other trace metal elements. These trace elements account for a very small proportion of a healthy human body, but their role is crucial. Although excessive metal elements can cause toxicity, the lack of metal elements can also cause various physiological disorders. These findings indicate that the micronutrients represented by metal elements play an indispensable role in the metabolism of cells and even the body [94]. Materials scientists have discovered that the doping of micronutrient elements can significantly improve the biological properties of 3D-printed bioceramic scaffolds and give them new functions. A scaffold covered with metallic elements will decompose and corrode in the body due to the effects of cells and the chemical environment and release metal ions with biological activity. These ions can enter the cell to activate related signal pathways to change cell behavior and ultimately promote tissue regeneration [88]. As the main component of drugs used to treat depression, the Li element is effective in treating osteoarthritis in recent years [95]. The underlying mechanism is that Li^+^ can activate autophagy to protect chondrocytes and tissues from inflammatory osteoarthritis [96]. Ma et al. [97] reported a Li-doped calcium polyphosphate (CPP) bioceramic scaffold. They found that the incorporation of the Li element can improve the biodegradability of the CPP scaffold. The doped CPP scaffold with a Li content of 2.0% is most conducive to cell proliferation and adhesion. This scaffold can upregulate the Wnt signaling pathway and increase the expression of the osteogenic marker alkaline phosphatase (ALP) when culturing MG63 cells. The deposition of calcium phosphate is conducive to the formation of a new bone in the body. In a recent study, the use of a bioceramic scaffold of calcium silicate (Li_2_Ca_4_Si_4_O_13_, LCS) doped with Li elements was tested [98]. The research found that the LCS scaffold can induce macrophages to transform to the anti-inflammatory M_2_ phenotype, while downregulating the expression of inflammatory factors TNF-α, IL-6, and IL-1β and upregulating the expression of anti-inflammatory factors IL-10 [98]. Moreover, the conditioned medium obtained by culturing macrophages on the LCS scaffold can promote the proliferation, migration, and maturation of chondrocytes. Sr is an alkaline earth metal element that exists in human bones. Sr^2+^ can increase bone density by promoting the activity of osteoblasts while inhibiting the function of osteoclasts [99]. This feature has a significant impact on increasing the use of Sr^2+^ in the treatment of osteoporosis on a large scale [100]. Zeng et al. prepared a Sr-doped calcium phosphate silicate (CPS) bioactive ceramic [101]. The extract of Sr-CPS can promote osteogenesis by upregulating the Wnt/β-catenin signaling pathway and inhibit osteoclastization by downregulating the NF-κB signaling pathway. Compared with CPS, Sr-CPS can significantly promote the repair of skull defects in osteoporotic animals. Lin et al. [102,103,104] systematically studied the improvement of Sr element doping on the properties of bioceramic materials. Studies have found that Sr^2+^ can promote the osteogenic differentiation of bone marrow mesenchymal stem cells and the angiogenic differentiation of human umbilical vein endothelial cells by activating the ERK/p38 signaling pathway [102,103,104]. In addition, Sr^2+^ can also regulate the transformation of macrophages into the anti-inflammatory M2 phenotype. This is useful for tissue regeneration.

Cu^2+^ is a heavy metal ion that can be used as a cofactor to participate in the normal metabolism of cells, such as mitochondrial respiration, enzyme redox, and free radical scavenging; but at the same time, excessive Cu^2+^ can inhibit the action of related proteases or destroy intracellular components [105]. Ai et al. prepared a Cu-doped hydroxyapatite scaffold through ion exchange and 3D printing technology. The Cu^2+^ in the scaffold can effectively kill bacteria, thereby giving the scaffold antibacterial activity. Whereas the 5Cu-HA scaffolds had higher mechanical properties and antibacterial properties than other non-doped Cu-HA scaffolds. At the same time, it showed lower cytotoxicity of bone marrow mesenchymal stem cells [106]. Baino F. studied the effect of Cu doping on the biological properties of mesoporous bioactive glass (MBG) scaffolds [107].

The results of in vitro cell experiments show that the Cu-MBG scaffold and its ion extract can stimulate human bone marrow mesenchymal stem cells to express the hypoxia-inducible factor (HIF-1α) and vascular endothelial growth factor (VEGF). This is conducive to the growth of new blood vessels in the body. Furthermore, the Cu-MBG scaffold and its ion extract can also promote the expression of osteogenic genes by stem cells. Lin et al. also prepared a Cu-doped bioactive glass-ceramic (BGC) scaffold through 3D printing technology [108]. The Cu-containing products released by it can promote the maturation of chondrocytes. The probable underlying mechanism is that Cu^2+^ can activate the HIF pathway and further enhance the conversion of macrophages to the M2 anti-inflammatory phenotype. Thus, the expression of anti-inflammatory factors is regulated, which will reduce cartilage tissue damage caused by inflammation. In addition to the mentioned Li, Sr, and Cu elements, Mn, Fe, Co, and other elements can also significantly improve the biological properties of 3D-printed bioceramic scaffolds. This confers special functions such as anti-bacterial and anti-tumor efficacy. This made it a multifunctional scaffold. In addition to being able to repair bone tissue defects, it is resistant to bacteria and tumors [109]. These studies show that elemental doping is an effective and very important scaffold modification strategy, and at the same time exhibits very promising clinical application value (Figure 6).

#### 3.1.3. Surface Functional Modification

In the development of biomaterials, various surface modification technologies represented by coatings have always played a very important role. Early inert alloy grafts could not be combined with human tissues well and aseptic loosening often occurred [110]. To improve the ability of the surface of the graft to bond with bone tissue, materials scientists try to spray bioactive materials on the surface of the graft. These bioactive coatings can effectively induce the combination of new bone and the graft, thereby appropriately extending the service life of the artificial graft. Surface modification technologies such as coating can also improve the surface performance of 3D-printed bioceramic scaffolds. Modification of the scaffold surface has a significant benefit by greatly improving the adhesion and proliferation of cells on the scaffold surface. At the same time, the functionally modified surface can enhance the mechanical properties of the scaffold or bring special functions such as antibacterial and photothermal effects to traditional 3D-printed bioceramic scaffolds [111,112,113,114].

The significant disadvantages of traditional 3D-printed bioceramic scaffolds are poor mechanical properties and brittleness. Sharifi et al. [115] immersed the HAP scaffold in two unsaturated derivatives of PCL fumarate (PCLF) and PCL itaconic acid (PCLI). In PCLI, a bioceramic scaffold with a polymer coating on the surface is obtained. The test results show that by changing the concentration of PCLF or PCLI solution, the compressive strength of the scaffold can be increased by 14% to 328%. In all samples, the mechanical enhancement effect of PCLF coating is better than that of PCLI. In addition, the polymer-modified HAP scaffold showed no obvious toxicity when culturing human primary osteosarcoma cells (G92 cell line). Although the introduction of polymer coating can significantly improve the mechanical properties of the scaffold, it cannot significantly promote cell adhesion and proliferation. To further enhance its biological properties, these polymer coatings need to be modified by additional grafting reactions. To enhance the compressive strength of the scaffold while improving the biological activity of the surface of the bioceramic scaffold, the research group coated the mesoporous bioactive glass on the β-TCP scaffold by spin coating [115]. Experiments show that the mesoporous bioactive glass nanolayer with a thickness of about 100 nm can significantly enhance the mechanical properties of the β-TCP scaffold and the deposition of apatite mineralization. In addition to releasing Ca^2+^ and PO4^3−^, the surface-modified scaffold can also release SiO_4_^4−^ in vitro and in vivo experiments. This enhances the expression of bone-related genes and proteins in rabbit bone marrow mesenchymal stem cells, and further promotes the expression of vascular-related genes in human umbilical vein endothelial cells.

In the surgical process of transplanting tissue engineering scaffolds, the risk of bacterial infection is extremely likely to lead to the failure of the operation. Therefore, materials scientists need to improve the bone reparability of the bioceramic scaffold while considering the antibacterial and sterilization ability of the scaffold. It was reported that a β-TCP scaffold was modified by Ag@GO nanocomposite prepared by immersion method [116]. The study found that the content of Ag nanoparticles on the surface of the scaffold can be adjusted by changing the number of dipping and the concentration of the Ag@GO nanocomposite solution. In the antibacterial experiment, the Ag@GO nanocomposite on the surface of the scaffold can effectively inhibit the proliferation of *E. coli*. Compared with the β-TCP scaffold, the modified scaffold not only has excellent antibacterial properties but also has a better bone-forming ability.

Traditional 3D-printed bioceramic scaffolds can only promote the repair of tissue defects but cannot treat tumors, which will limit the clinical application of 3D-printed bioceramic scaffolds. To give scaffolds the ability to treat tumors, Zhang et al. modified graphene oxide (GO) as a coating on β-TCP support by a simple dipping method (Figure 4) [117]. Research has found that the GO-TCP scaffold can exhibit excellent photothermal effects even when the optical power is as low as 0.36 W/cm^2^ under the irradiation of 808 nm near-infrared light. In addition to killing tumor cells around the defect site, the GO-TCP has a better bone-promoting ability than pure TCP scaffolds and can upregulate the level of osteogenesis-related genes expressed by rabbit bone marrow mesenchymal stem cells. However, the biodegradability and long-term toxicity of GO have been controversial since it was used as a biological material. To avoid the biological toxicity of GO to cells and expand the photothermal therapy in the field of tissue engineering. Ma et al. also reported a bifunctional 3D-printed bioceramic scaffold with a polydopamine nano-coating on the surface [118]. The polydopamine on the surface gives the scaffold good biocompatibility, biodegradability, and excellent biodegradability. The photothermal performance makes the scaffold not only promote bone defect repair but also eliminate cancer cells remaining at the defect site after tumor excision.

### 3.2. Improvement in the Material Structure

When designing 3D-printed bioceramic scaffolds, materials scientists not only need to consider the composition of the material, but also need to pay attention to the role of the structure of the scaffold in tissue repair. With the development of tissue engineering, materials scientists have discovered the design of the pore size and structure in the scaffold. The micro-nano structure on the surface of the scaffold and various bionic structures plays an important role in the process of osteogenesis and vascularization of bone tissue defects. When preparing porous bioceramic scaffolds, 3D printing technology and other additive manufacturing technologies have unprecedented advantages compared with traditional processes such as the pore former method, freeze-drying method, and organic foam template method [119]. 3D printing technology represented by inkjet 3D printing technology, selective laser sintering technology, ink direct writing 3D printing technology, and SLA printing technology can produce more complex, finer, and more customized three-dimensional porous scaffolds. This provides a more operational platform for basic research in tissue engineering, and more possibilities for clinical solutions in regenerative medicine [120,121].

#### 3.2.1. Optimizing the Porous Structure

The porous structure of the tissue engineering scaffold plays a vital role in the transmission and exchange of nutrients and the growth of new bone tissue and new blood vessels [122]. The pore size and porosity will affect the behavior of cells and even the process of tissue regeneration [123]. For example, when the scaffold pore size is small, it will inhibit the growth of new bone and blood vessels and cause the formation of fibrotic tissue, while the porosity of the scaffold with a larger pore size is conducive to vascularization and new bone formation in vivo [124]. As early as 1996, Tsuruga et al. [125] studied the effects of hydroxyapatites with pore sizes (106–212, 212–300, 300–400, 400–500, and 500–600 µm) in rat subcutaneous tissues to enhance bone-forming capacity. If we compare the activity of alkaline phosphatase with the content of osteocalcin, we find that materials with a pore size of 300–400 μm have the best osteoclastogenic capacity. This indicates that materials with a pore size of 300–400 μm may be more useful for promoting osteoblast adhesion, proliferation, and angiogenesis at the defect site. Entezari et al. [126] printed bioceramic scaffolds with different porosities and pore structures for bone defect repair experiments, and their results also confirmed that scaffolds with a pore size of about 390 μm have the best bone formation effect in vivo. However, when the size is greater than 590 μm, the change in the pore size of the scaffold will not bring any improvement to the formation of new bone.

In addition to the size, the geometry of the pore structure also affects the cells. Rumpler et al. [127] prepared HAP ceramic sheets of different specifications including four kinds of channel shapes—triangle, square, hexagon, and round—and three kinds of channel sizes with lengths of 3.14, 4.71, and 6.28 mm respectively. Cell experiments have found that the surface area and local curvature of the scaffold can significantly affect the rate of tissue growth, and the amount of tissue deposition is proportional to the local curvature. This indicated that a single cell could perceive the difference in the surface morphology of the material, and the curvature of the surface of the material could drive the tissue growth. The above studies show that the porous structure of the material can not only facilitate the exchange of nutrients but also affect the behavior of cells. These phenomena mean that when designing and preparing 3D-printed bioceramic scaffolds, materials scientists need to purposefully optimize the porous structure of the scaffold to maximize tissue regeneration.

#### 3.2.2. Construction of Micro-Nano Structures

When tissue engineering scaffolds are transplanted into the body, they first encounter a variety of blood cells and immune cells. The surface properties of the scaffold will greatly affect the behavior of these cells, which in turn affects subsequent tissue regeneration and vascularization. In particular, materials with special micro/nano-scale morphologies on the surface can affect the adhesion of cells by promoting the adsorption of more specific proteins (Figure 7). This in turn causes changes in cell morphology and intracellular tension and ultimately changes the behavior and fate of cells by activating related intracellular signaling pathways [128,129,130,131,132]. These phenomena provide materials scientists with a unique way of modifying materials, revealing that the micro-nano-scale morphology plays an important role in the design and preparation of the scaffold. Xiao et al. [133] constructed different nanostructures on HAP scaffolds by hydrothermal method assisted by small-molecule 1,2,3,4,5,6-cyclohexanehexacarboxylic acid (H6L). Experiments show that as the concentration of H6L molecules increases, the microstructure on the surface of the HAP scaffold gradually changes from a plate shape to a linear shape, and finally to a spherical shape. Compared with other groups, scaffolds with spherical nanostructures in the surface morphology are most conducive to the proliferation and osteogenic differentiation of mesenchymal stem cells in in vitro experiments. Not only small organic molecules can affect the results of hydrothermal products on the surface of the scaffold, but solutions containing metal ions can also affect the surface morphology of the scaffold in the hydrothermal method.

Elrayah et al. soaked the porous HAP scaffold in a copper-containing solution and then constructed different micro/nanostructures on the surface of the scaffold by hydrothermal method [134]. The study found that the concentration of Cu^2+^ can affect the final morphology of the surface microstructure of the scaffold as the concentration of Cu^2+^ increases; the surface of the scaffold gradually changes from a spherical structure to a flower-like structure. In in vitro experiments, the scaffold with a flower-like structure on the surface can significantly enhance the proliferation of endothelial cells compared to other scaffolds. Correspondingly, the scaffold with a flower-like surface in the in vivo experiment has the best vascularization effect. Similarly, Xia et al. [135] used PMMA (polymethylmethacrylate) microspheres as a template to prepare porous HAP and β-TCP scaffolds by injection molding, and then grew nanosheets, nanorods, and micro-nano composite (hybrid nanorods and micro rods) structure. Studies have found that these bioceramic scaffolds with micro/nanostructures on the surface can significantly promote the adhesion, proliferation, and ALP activity of bone marrow mesenchymal stem cells, and can upregulate the expression of osteogenesis-related genes. Moreover, these scaffolds can activate ERK (extracellular signal-related kinases) and p38 MAPK (mitogen-activated protein kinase) signaling pathways in bone marrow mesenchymal stem cells, thereby promoting stem cell spreading and osteogenic differentiation. The scaffold with a micro-nano composite structure on the surface has the best promotion effect.

In addition to repairing bone defects, scaffolds with special micro-nano morphology can also promote the regeneration of osteochondral defects. Deng et al. [136] reported that a bioceramic scaffold with a micro/nanostructure with a controlled morphology on the surface prepared by a hydrothermal method was used for the regeneration of cartilage and subchondral bone. In the experiment, the change of hydrothermal time and solution concentration can lead to different micro/nanostructures (nanoparticles, nanoflakes, and micro rods) on the surface of the final scaffold. These micro-nano-level calcium phosphate crystals can fill the cracks and gaps on the scaffold pillars to significantly enhance the compressive strength of the scaffold. Along with improving the mechanical properties of the scaffold, the micro/nanostructure on the surface of the scaffold can also regulate cell morphology and promote the spreading and differentiation of chondrocytes by activating integrin αvβ1 and α5β1 heterodimers. As well as through the synergy of integrin α5β1 and RhoA, it promotes the osteogenic differentiation of rabbit bone marrow mesenchymal stem cells. The scaffold with micrometer rods on the surface has the most obvious effect on the differentiation of chondrocytes and mesenchymal stem cells.

#### 3.2.3. Constructing a Bionic Structure

After epochs of evolution, organisms in nature have developed a variety of organs and tissues in different forms. These organs or tissues serve as special ‘weapons’ to help organisms, adapt to the environment, and thrive. With the development of science and technology, people use their ingenuity to invent tools, but they also continue to learn from the animals and plants in nature. Bionics attempts to analyze biological processes and implement them using advanced technology. The term ‘bionics’—combining the words ‘biology’ and ‘technology’—was coined by Jack Steele in the 1960s [137], and it translates the information processing ability to live systems into design challenges. It is the development of a set of functions based on a similar system found in nature [138]. Whereas biomimetics is essentially the process of mimicking the structure or function of a biologically produced substance or material to manufacture a synthetic product [137,138,139]. Through in-depth research and creative imitation, various powerful bionic materials have been discovered. For example, the unique surface micron-level structure of the lotus leaf has super-hydrophobic characteristics so that the water droplets slide off the leaf surface to take away the sludge and realize the self-cleaning function [140], and a material has been designed to imitate the surface microstructure of the lotus leaf that also has this hydrophobic function [141]. Another example is the organic–inorganic composite system similar to the “brick + mud” layered structure that makes shells have both high strength and high toughness, which also enables them to withstand the long-term erosion of sea waves without being destroyed [142].

**Figure 7 life-12-00903-f007:**
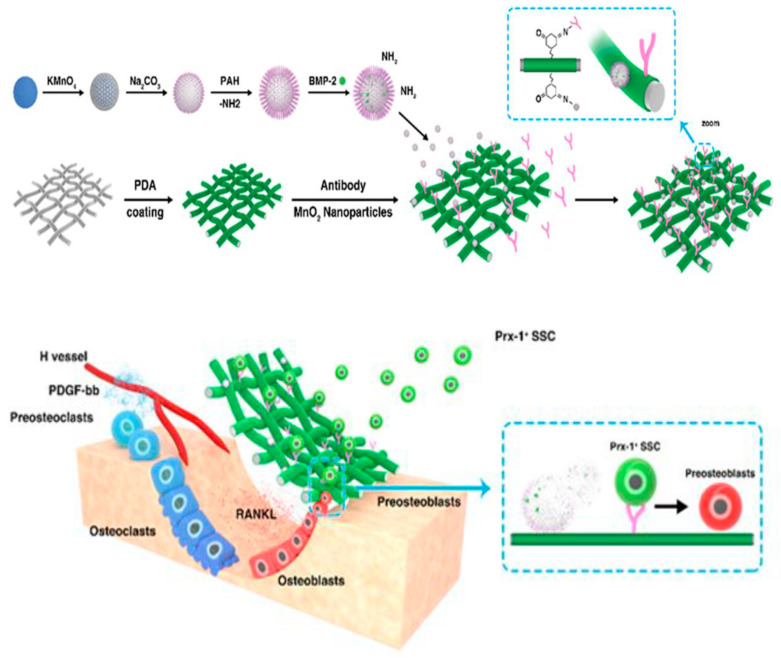
Constructing a bionic periosteum: a schematic diagram illustrating the process of preparing the electronic structure and how this electronic periosteum can stimulate the microenvironment at the site of bone regeneration. Notes: Reproduced with permission from Ref. [139]. Copyright 2020 ACS Publications.

3D printing materials also have excellent mechanical properties [143]. These bionic design ideas make the emergence of more high-performance materials possible. The structure of bionic natural bone tissue is one of the effective ideas for designing bone tissue engineering scaffolds. Meng et al. reported a bio-ceramic scaffold with a bionic Haversian bone structure prepared by DLP light-curing 3D printing technology [144]. By modifying the parameters of the multi-level structure in the scaffold model (such as the number of Haversian tubes) through the CAD software, the compressive strength and porosity of the bionic scaffold can be precisely controlled. In this study, a variety of cells (including bone marrow mesenchymal stem cells, endothelial cells, and Schwann cells) were seeded on the scaffold, which simulated the complex environment of bone tissue in the body. Through in vivo and in vitro experimental data, it is found that this bioceramic scaffold with a bionic Haversian structure can transport a variety of cells well and is beneficial to various cell interactions: it can induce osteogenic differentiation, vascularization, and vascularization of related cells in vitro. Neutralization in the body can accelerate the growth of blood vessels and the formation of new bones under the action of a variety of loaded cells. In addition to directly mimicking natural bone tissue, bionic materials inspired by other animals and plants can also give bioceramic scaffolds more excellent performance. Feng [145] and Zhang [146] et al. printed multi-channel support of bionic lotus root by improving the 3D printing nozzle. By increasing or decreasing the number of parallel needles in the printing nozzle, the number of hollow tubes in a single pillar of the printed scaffold can be adjusted. In vitro experiments show that this kind of scaffold can induce the migration of endothelial cells to facilitate vascularization in vivo, and it can also transport stem cells and growth factors to further promote tissue regeneration. In vivo experiments show that this kind of bionic lotus root porous channel structure scaffold can significantly promote the growth of blood vessels and the formation of new bone after being transplanted into the body. Based on the bionic lotus root multi-channel scaffold, Li et al. [147] also reported a 3D-printed bioceramic scaffold for drug loading and cell transport inspired by the hot dog structure. This study combined the extrusion 3D printing technology with the two-way ice template method to prepare a bioceramic rod with both a hollow bioceramic tube (imitating the bread structure in a hot dog, with a tube diameter of about 1 mm) and a hot dog intestine shape (imitating the sausage structure in a hot dog, with a diameter of about 500 μm; the stick has a uniform.

The arranged layered micropore structure, layered micropore diameter is about 30 μm multilayer structure scaffold. The hollow tube structure of the scaffold facilitates the growth of blood vessels and new bone tissue. The ‘sausage’ structure in the scaffold can be made into bioceramic rods of different materials by changing the solution used and can carry different drugs or proteins according to different needs. In vivo and in vitro experiments show that this kind of bionic hot dog scaffold can load and release drugs and proteins well, and it can also promote the differentiation and transport of tissue cells. This bionic scaffold has good performance in drug delivery, tissue engineering, and regenerative medicine.

## 4. Conclusions and Prospects

Bone tissue has a good self-healing ability. Despite this, congenital and acquired diseases—including trauma, infection, and tumors—can make a patient’s bone defect exceeds the critical size of what the body can fill and is therefore incurable. Clinical bone defect treatment modalities have certain therapeutic effects, albeit they also have various limitations that cannot be ignored. Therefore, tissue engineering scaffolds represented by bioceramic scaffolds came into being. The traditional bioceramic scaffold has a simple structure, single function, and unsatisfactory mechanical and biological properties. After years of research, the bioceramic scaffold has been greatly improved. On the other hand, materials scientists begin with the development of the material, by doping with micronutrients and modifying the functional surface. Thus, giving conventional scaffolds better biological and mechanical performance, as well as developing additional antibacterial and antitumor properties and other functions. Furthermore, researchers are exploring optimal solutions for the 3D printing of bioceramic scaffolds. Starting from the material structure through optimizing the porous structure of the scaffold, building micro/nanostructures on the surface, and building a multilayer electronic structure to enrich and explore the diversity of the scaffold structure. These fruitful improvements make 3D-printed bioceramic scaffolds show appropriate potential for clinical applications and provide a very promising solution to the medical problem of bone defects. Table 4 briefly summarizes the challenges and suggests possible solutions for 3D printing of bioactive ceramics.

Although existing research has significantly improved the performance of traditional scaffolds, 3D-printed bioceramic scaffolds are still far from clinical applications and there are also many challenges and problems. Firstly, the existing 3D printing technology is difficult to prepare a bioceramic scaffold with both high strength and good toughness. Researchers generally obtain high-strength porous bioceramic scaffolds through high-temperature sintering. However, since this pure ceramic scaffold only contains the ceramic phase, there are problems of insufficient toughness and easy fracture during actual use. These problems make 3D-printed bioceramic scaffolds unable to adapt to the special mechanical environment of load-bearing bones like metal implants, thus limiting the application range of ceramic scaffolds. In subsequent research, materials scientists can improve the existing 3D printing technology and imitate the composition and structure of natural high-strength and high-toughness materials (bones, mussels) to prepare composite multi-material 3D with excellent mechanical properties to print the bioceramic scaffold so that it can be used to repair bone defects in load-bearing parts. Secondly, in clinical practice, some cases need to treat the disease and repair the bone defect at the same time. For example, patients with bone tumors leave a large bone defect after the tumor is removed, and there are residual tumor cells around the defect site. This complicated situation requires bone repair materials that can kill residual tumor cells to prevent cancer recurrence and promote the growth of new bone tissue, blood vessels, and nerves to restore bone structure and function to the maximum extent. To meet these harsh conditions, materials scientists need to make full use of the advantages of 3D printing technology and compound a variety of functional materials based on an in-depth study of the composition and structure of natural bone tissue, to develop materials that can be used for both disease treatment and 3D printed multifunctional bioceramic scaffold for tissue regeneration. Thirdly, the existing 3D-printed bioceramic scaffolds are difficult to accurately imitate the highly complex and ordered microstructure of raw natural bone tissue. Some scholars have divided the structure of bone into 12 levels from the macro-level to the micro molecular level [10].

At present, most of the bionic designs on bone tissues are at the level of biomimetic compact bone and cancellous bone. There are few reports in the literature on the realization of the multi-level structure of bionic bone from nanoscopic to microscopic to macroscopic. To start repairing bone tissue at the nanometer scale, materials scientists may need to try to integrate other micro-nano manufacturing technologies—such as hydrothermal processes, laser engraving technology, and electrospinning technology—into the existing 3D printing technology. A scaffold with a finer structure is produced, to realize the regeneration of the fine structure of bone tissue. Lastly, natural bone tissue is composed of multiple cells and substances. It plays a vital role in supporting the body, protecting internal organs, and hematopoiesis, and acting as a mineral reserve [148]. The existing 3D-printed bioceramic scaffolds can only fill the defect and promote the growth of new bone and blood vessels, but they cannot restore the full functions of the bone tissue. Materials scientists need to use multi-channel 3D printing technology to combine a variety of materials and a variety of cells to simulate the real situation of bone tissue in the body as much as possible, to achieve not only the structure but also the function of the bone tissue. With the rapid development of modern technology, the emergence of various advanced manufacturing technologies has made the form of biomaterials and their functions more diversified, and their composition and structure more refined. The vigorous development of computer technology has also laid a solid foundation for the customized design and precision medicine. In the future, with the in-depth cross integration of multiple materials and multiple technologies, the 3D-printed bioceramic scaffolds used in the field of regenerative medicine will inevitably develop by leaps and bounds. The vision here has a wide scope, many challenges, and a long road ahead.

## Figures and Tables

**Figure 1 life-12-00903-f001:**
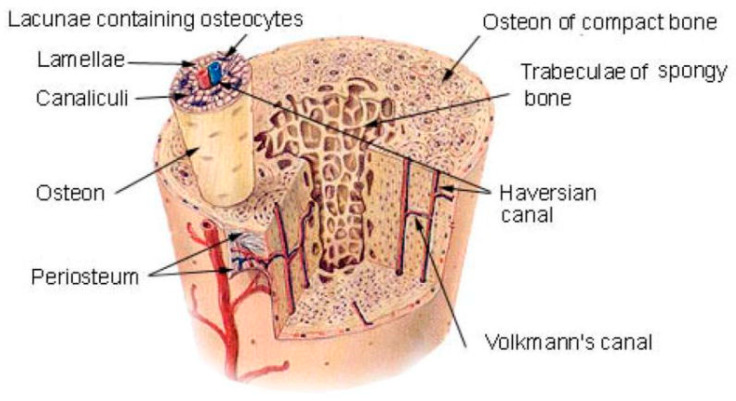
Human bone tissue is characterized by a complex and highly organized hierarchical structure. Notes: Reproduced with permission from Ref. [15]. Copyright 2010 University of Virginia ProQuest Dissertations Publishing.

**Figure 2 life-12-00903-f002:**
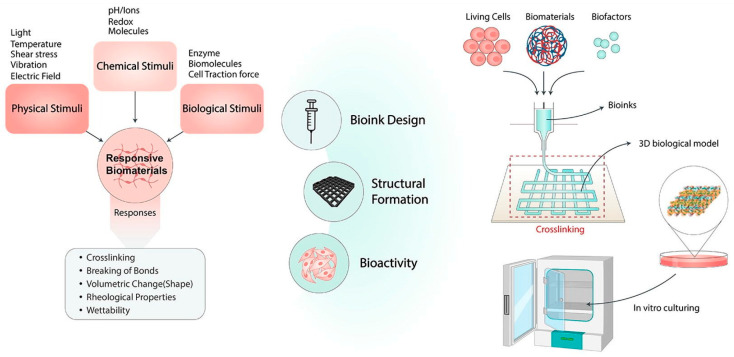
Schematic diagram of application in 3D bioprinting. Notes: Reproduced with permission from Ref. [23]. Copyright 2022 Elsevier.

**Figure 3 life-12-00903-f003:**
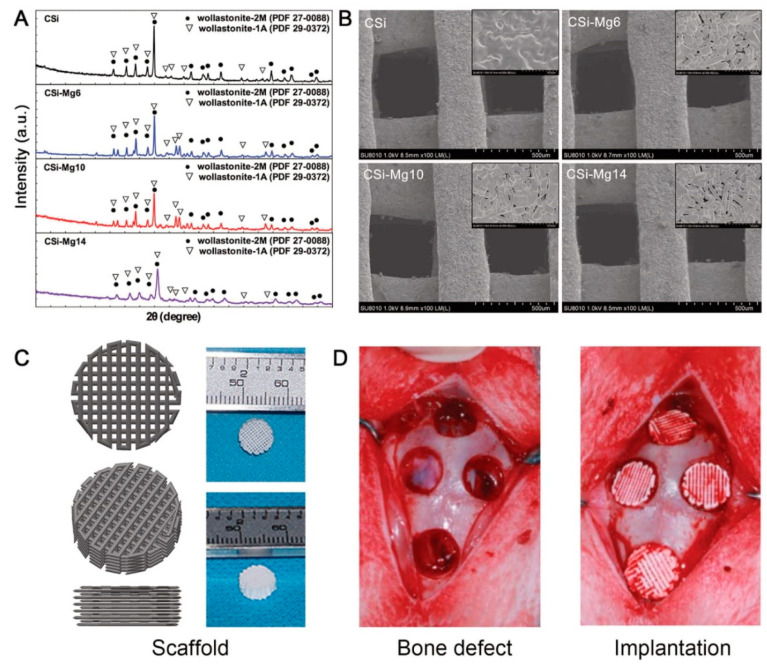
Characterization of 3D printing ceramic powder and support (**A**) XRD pattern of ceramic powder. (**B**) SEM images of surface morphology and microstructure of ceramic supports. (**C**) 3D models and macro drawings of representative ceramic scaffolds. (**D**) Implantation of bone defects and ceramic scaffolds in rabbit skull defects. Notes: Reproduced with permission from Ref. [47]. Copyright 2016 Scientific Reports.

**Figure 4 life-12-00903-f004:**
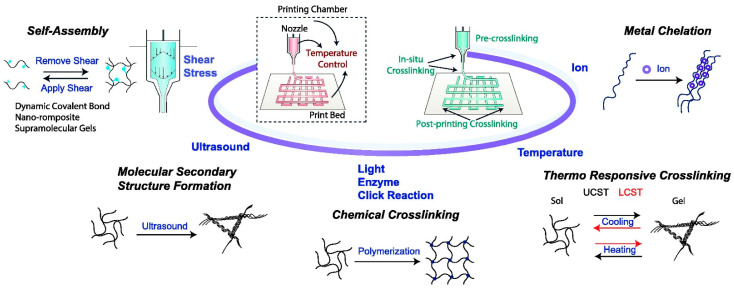
Bioink crosslinking mechanisms and application strategies in extrusion-based 3D bioprinting. The stimulus can be applied to the printing chamber, nozzle, or print bed. Crosslinking can take place before, in situ, and after printing. Reproduced with permission from Ref. [23]. Copyright 2022 Elsevier.

**Figure 5 life-12-00903-f005:**
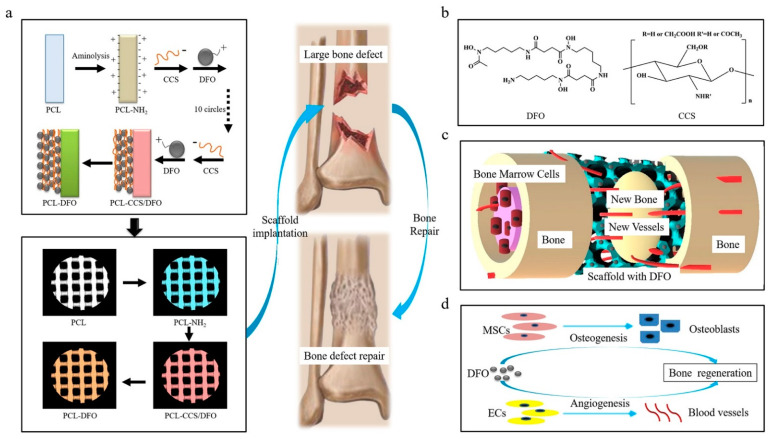
Schematic diagram of bridging deferoxamine (DFO) on the surface of 3D printed polycaprolactone (PCL) scaffold and its biological function for bone regeneration in bone defect model. (**a**) Up panel: Diagram showing the preparation process of PCL-DFO scaffolds including surface aminolysis and layer-by-layer assembly with oppositely charged carboxymethyl chitosan (CCS). Lower panel: Four scaffolds were used in animal study including the pure PCL, their intermediate product PCL-NH2, and the final product PCL-DFO. (**b**) The chemical molecular structure of DFO (left) and CCS (right). (**c**) Schematic diagram showing the effect of PCL-DFO scaffold on angiogenesis and osteogenesis at the bone defect site. (**d**) The cellular mechanism of promoting bone regeneration by DFO in mesenchymal stem cells (MSCs) and in vascular endothelia cells (ECs). Reproduced with permission from Ref. [14]. Copyright 2019 Elsevier.

**Figure 6 life-12-00903-f006:**
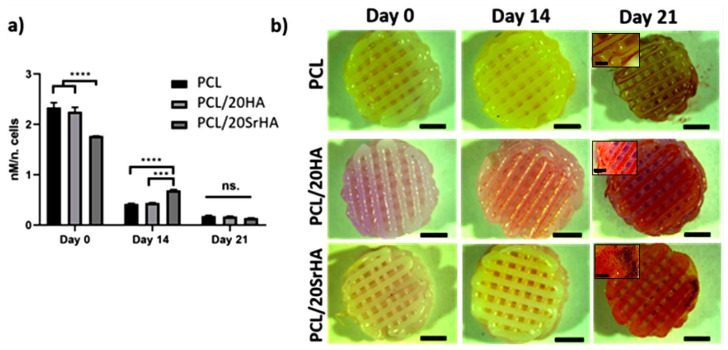
(**a**) Osteogenic differentiation of hTERT MSCs seeded on the strontium-doped 3D printing scaffolds through measurement of ALP activity up to 21 days. **** *p* < 0.0001, *** *p* < 0.001, the ns is short for no significance. (**b**) Alizarin red staining of cell-seeded scaffolds. Notes: Reproduced with permission from Ref. [104]. Copyright 2020 Elsevier.

**Table 1 life-12-00903-t001:** Summary of some biomaterials used for orthopedic applications.

Types	Biomaterials	Advantages	Disadvantages	Composite Materials	In Vitro Study	In Vivo Study	Reference
Natural polymer materials	Chitosan	Excellent biocompatibility, osteogenic potential, compatibility, cytocompatibility.	Strong biodegradability, fast degradation speed, easy to deform	Chitosan-based SiO_2_ nanocomposites	Human osteoblasts (HOBs)s were used to detect cell adhesion and proliferation of scaffolds.	Scaffolds were implanted in nude mice to verify osteogenesis and vascularization.	[36,37]
	Alginate	Excellent biocompatibility, biodegradability, hydrophilicity, and low cost can be shaped.	Poor bioactivity, antioxidant, mechanical strength, and bone conductivity.	Alginate microbeads (AM) loaded with BMP-2.	Active expression of ALP in mesenchymal stem cells was used to examine the release of alginate microbeads carrier BMP-2.	Skull defect model rats and mice were injected subcutaneously to verify the higher osteogenic efficiency of alginate microbeads carrier BMP-2.	[38]
	Collagen	Excellent biocompatibility and biodegradability; easily degrades and strong plasticity and low immunogenicity.	Fast degradation rate and poor mechanical properties	Mineralized collagen-hydroxyapatite-based scaffolds	Mouse calvarial 3T3 (MC3T3) cells were used to examine the in vitro cytocompatibility of various scaffolds. Osteogenic differentiation with fluorescent multi reporter mice BMSCs.	A mouse skull defect model was used to observe the bone regeneration ability of different scaffolds in vivo.	[39,40]
Artificial synthetic materials	Polylactic acid	Good biodegradability, biocompatibility, and processability; high mechanical strength.	Slow degradation rate, poor osteoconductivity.	Tantalum-coated polylactic acid fibrous membranes.	Preosteoblast cell lines (MC3T3-E1) were used to verify the biocompatibility of Ta-PLA electrospun membranes.	Rabbits with cylindrical skull defects were used to examine the osteogenic effect of Ta-PLA electrospun membranes.	[41]
	Polycaprolactone (PCL)	Good biocompatibility, biodegradability, and processability.	Poor bioactivity, Slow degradation rate, and long degradation cycle.	Polycaprolactone/ chitosan-g-polycaprolactone/hydroxyapatite electrospun nanocomposite scaffolds.	NIH 3T3 fibroblast cells and MG-63 cells were used to study the in vitro cytocompatibility of nanocomposite scaffolds.	PCL implantation in bone defect mice can promote bone defect repair with good cellular compatibility.	[42,35]

**Table 2 life-12-00903-t002:** Various characteristics of bioceramic materials with advantages and disadvantages.

Bioceramic Materials	Characteristic	Advantages	Disadvantages	Products	Reference
Alumina	Alumina is an inert ceramic material with good chemical stability and high mechanical strength. Abundant raw materials, low price, wide use, high mechanical strength, pressure resistance, high-temperature resistance, corrosion resistance, high-temperature insulation, and excellent dielectric properties.	Stability, biocompatibility, and excellent wear resistance, non-cytotoxic.	Limited strength, low mechanical properties.	Inert alumina ceramics, nanoporous alumina.	[47,48,49]
Zirconia	Similar to inkjet 3D printing, a liquid binder is used to bind the powder together and then the support layer is printed layer by layer, finally, the powder printing stand is melted directly. High mechanical strength, high strength, high toughness, high hardness, excellent chemical corrosion and wear resistance, low thermal conductivity, good insulation, and self-lubrication.	Fracture resistance and flexural strength characteristics.	Micro-cracks or inducing a phase transformation (grind or sandblasting dental treatment), Chemical aging, and wear.	Yttria-stabilized tetragonal zirconia polycrystalline (Y-TZP), zirconias versus silica-based ceramics.	[50,51]
Bioactive glass	Bioactive glass exhibits uniform interconnected macro-pores, high porosity, and high compressive strength. It can promote the expression of osteogenic genes in human bone marrow stromal cells. High biological activity, osteogenesis, osteoinduction, good combination with bone and soft tissue, and many functions.	Good bioactivity, biocompatibility, and no cytotoxicity promote bone and soft tissue regeneration.	Poor mechanical strength and intrinsic brittleness.	Bioactive glass ink; bioactive borosilicate glass (BG) scaffolds.	[52,53]
Glass-ceramics	Glass-ceramics are mainly composed of ~70 vol % of interlocked rod-like lithium disilicate crystals with high compressive strength. High mechanical strength, adjustable thermal expansion, chemical corrosion resistance, and wide application.	It has sufficient strength and chemical stability, with outstanding aesthetics, transparency, as well as low thermal conductivity with adequate strength. In addition to biocompatibility, corrosion resistance, and chemical durability.	The production process is complicated and high cost.	Strontium doping glass-ceramic material, TiO_2_-containing glass-ceramics.	[54,55]
Hydroxyapatite	Principal inorganic component of human or animal bones and teeth.	Good biocompatibility, bioactivity, and bone conductivity.	The degradation rate is slow, has a poor bone induction effect, and has high brittleness.	Hydroxyapatite coatings, poly (glycolic acid)/hydroxyapatite composite scaffolds.	[56,57,58]
Calcium phosphates	Similar in composition to bone minerals, the most widely used synthetic bone substitutes.	Excellent biocompatibility, bioactivity, bone conductivity, and absorbability.	Low compressive strength, no toughness, slow degradation.	Beta-tricalcium phosphate (β-TCP)-based bioinks, 3D printed calcium phosphate cement (CPC).	[59,60,61]

**Table 3 life-12-00903-t003:** A brief listing of the most used 3D printing technologies.

3D Printing Technologies	Principle	Advantages	Disadvantages	Reference
Inkjet 3D printing technology	The print head sprays an adhesive over a specific area to bind the powder material together, then accumulates layer by layer to form the final scaffold frame.	Low cost, a wide range of applications, printing does not require additional support.	The mechanical properties of the scaffold are low, the surface is very rough, and poor printing accuracy.	[9,10,11,12]
Selective laser sintering technology	Similar to inkjet 3D printing, a liquid binder is used to bind the powder together and then the support layer is printed layer by layer, finally, the powder printing stand is melted directly.	No additional support is required, printed metal material.	High cost, low efficiency, the rough surface of the scaffold, low resolution, and long printing time.	[13,14,15]
Ink direct writing 3D printing technology	The mobile print head directly extrudes the printing ink layer by layer to build a three-dimensional scaffold.	Fast printing speed, easy operation, low cost, good printing accuracy, widely used.	Low printing accuracy, additional support is needed to assist with printing, sag and deformation may occur.	[16,17,18]
SLA printing technology	The 3D scaffold is printed layer by layer through photoinduced polymerization of photosensitive resin.	High accuracy allows printing of scaffolds with complex porous structures and very high resolution.	Need additional support, post-cleaning takes a lot of time and energy and affects roughness.	[19,20]

**Table 4 life-12-00903-t004:** Challenges and possible solutions for 3D printing bioactive ceramics.

Challenges	Solutions
Existing bioceramic scaffolds have insufficient toughness and are easy to fracture, so they cannot be used for bearing bones.	3D printing technology and bionic technology to prepare composite multi-materials, with excellent mechanical properties of 3D-printed bioceramic scaffold.
Clinical practice often requires the simultaneous treatment of the patient’s disease and repair of bone defects.	3D printing technology combined with drug-carrying materials and bone growth-promoting factors has developed a 3D-printed multifunctional bioceramic scaffold that can be used for both disease treatment and tissue regeneration. The scaffolds can both treat disease and promote bone tissue regeneration.
Existing 3D-printed bioceramics scaffolds are difficult to accurately mimic the highly complex and ordered microstructure of natural bone tissue.	Other micro-nano manufacturing technologies—such as hydrothermal processing, laser engraving, and electrospinning—are being combined with existing 3D printing technologies to produce scaffolds with finer structures.
Existing 3D-printed bioceramic scaffolds cannot restore the full function of bone tissue.	Through the multi-channel 3D printing technology, a variety of materials and cells are combined to simulate the real situation of bone tissue in the body as much as possible.
Existing 3D printing technology is difficult to be accurate to the nanometer scale, and can only be made into a scaffold and change its shape through physical and chemical methods.	The development of nano-scale 3D printing technology can prepare multi-tissue scaffolds with spatial and functional regulation.

## Data Availability

The datasets used and/or analyzed during this study are available from the corresponding author on reasonable request.

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
