# Peer review of "Bone Tissue Engineering through 3D Bioprinting of Bioceramic Scaffolds: A Review and Update"

_life, 2022, doi:10.3390/life12060903_

Round 1

Reviewer 1 Report

The manuscript has been organized in a logical order with adequate discussion. The manuscript also is well-written that can contribute to the field. I think the manuscript can be considered for publication after minor revisions:

1) the quality of images is poor. Please try to improve the quality of the figure.

2) Please include a table summarizing the current challenges and possible solutions. 

3) Please shorten the section related to the different bioprinting technology as a lot of review papers have already covered those.  

4) Please highlight the significance of the current review in the abstract properly and avoid presenting general information. 

Author Response

  1. Reviewer 1

Comments and Suggestions for Authors

The manuscript has been organized in a logical order with adequate discussion. The manuscript also is well-written that can contribute to the field. I think the manuscript can be considered for publication after minor revisions:

Dear Sir,

Thank you very much for your nice comments on our article.

  • the quality of images is poor. Please try to improve the quality of the figure.

-Done

2) Please include a table summarizing the current challenges and possible solutions. 

-Done, please see table 4

  • Please shorten the section related to the different bioprinting technology as a lot of review papers have already covered those.  

-Done, some sentences have been deleted and the paragraph was shortened without compromising the meaning

  • Please highlight the significance of the current review in the abstract properly and avoid presenting general information. 

-Thanks for the valuable note. It was done

Reviewer 2 Report

This review paper is very comprehensive and well-suited for publication in life. The authors wrote a comprehensive and well-structured on 3D bioprinting of bioceramic scaffolds for applications in bone tissue engineering. It covers all the important areas of this topic such as biomaterials, 3DP technologies, and recent improvements in the field. The only point I would suggest is to add a figure to visually represent section 3.2.3 on “Constructing a bionic structure”. Also, what does ‘bionic’ mean here? How does this differ from biomimetic structures or features? Please expand on this.

Author Response

  1. Reviewer 2

Comments and Suggestions for Authors

This review paper is very comprehensive and well-suited for publication in life. The authors wrote a comprehensive and well-structured on 3D bioprinting of bioceramic scaffolds for applications in bone tissue engineering. It covers all the important areas of this topic such as biomaterials, 3DP technologies, and recent improvements in the field.

The only point I would suggest is to add a figure to visually represent section 3.2.3 on “Constructing a bionic structure”. Also, what does ‘bionic’ mean here? How does this differ from biomimetic structures or features? Please expand on this.

Dear Sir,

Thank you very much for your nice comments on our article.

The suggested figure was added.

We have expanded on the ‘bionic’ explanation and how does this differ from biomimetic structures or features in that section.

Thanks

Reviewer 3 Report

In this article, authors summarized development of bone tissue engineering scaffold using various 3D printing technology. Especially, as printing materials, authors focused on the bio-ceramic materials. They updated current research results related bio-ceramic scaffolds and methods for improving the performance of bio-ceramic scaffolds. This article was well organized and sentences were cleared. However, several contents were sparse about current research results. If contents are reinforced in consideration to the following comments, this article will be published to the ‘Life’.   

1. Please indicate the source of Figure 1 and attach a reference.

2. In the case of PCL, many in vitro and in vivo studies for bone tissue reconstruction have been conducted. Various research results and references related to the PCL in vivo study in ‘Table 1’ should be described.

For example:

Macromol Biosci. 2021 Dec;21(12):e2100266. doi: 10.1002/mabi.202100266. Epub 2021 Oct 13.

Biomed Mater. 2020 Jul 1;15(4):045022. doi: 10.1088/1748-605X/ab843f.

Biomater Sci. 2021 Jun 7;9(11):4019-4039. doi: 10.1039/d1bm00062d. Epub 2021 Apr 26.

Acta Biomater. 2019 May;90:393-402. doi: 10.1016/j.actbio.2019.04.019. Epub 2019 Apr 6.

Acta Biomater. 2016 Jan;30:319-333. doi: 10.1016/j.actbio.2015.11.012. Epub 2015 Nov 10.

J Orthop Res. 2020 May;38(5):961-971. doi: 10.1002/jor.24542. Epub 2019 Dec 5.

Macromol Biosci. 2021 Mar;21(3):e2000336. doi: 10.1002/mabi.202000336. Epub 2020 Dec 21.

Materials (Basel). 2018 Feb 4;11(2):238. doi: 10.3390/ma11020238.

Int J Mol Sci. 2021 Mar 30;22(7):3588. doi: 10.3390/ijms22073588.

Gels 2022 Mar, 8:163. Doi:10.3390/gels8030163

Tissue Eng Part A. 2011 Jul;17(13-14):1831-9. doi: 10.1089/ten.TEA.2010.0560. Epub 2011 Apr 27.

Please search ‘pubmed.gov’ website.

Since there are many research cases on in addition and other materials (including chitosan, alginate, collagen, PLA), references and related information should be added.

3. In table 3, advantage section of bioactive glass and glass-ceramic was combined. Even if the content is the same, please write the core content separately for each bioceramic material.

4. There is also a selective laser melting technology that melts powder by using a higher-power laser, although it is similar to SLS (Selective laser sintering). At the end of ‘2.3.2. Selective laser sintering technology’ section, please also investigate and describe the SLM technology.

5. Fused deposition modeling (FDM), the most widely used 3D printing, that is a 3D printing method based on a continuous filament extrusion was not reported. Additional section of FDM is required.

6. Change the title of 2.3.3 to ‘Direct ink writing 3D printing technology’ in the same way as the contents.

Author Response

Reviewer 3

Comments and Suggestions for Authors

In this article, authors summarized development of bone tissue engineering scaffold using various 3D printing technology. Especially, as printing materials, authors focused on the bio-ceramic materials. They updated current research results related bio-ceramic scaffolds and methods for improving the performance of bio-ceramic scaffolds. This article was well organized and sentences were cleared. However, several contents were sparse about current research results. If contents are reinforced in consideration to the following comments, this article will be published to the ‘Life’.   

Dear Sir,

Thank you very much for your nice comments on our article.

  1. Please indicate the source of Figure 1 and attach a reference.

 - Done

  1. In the case of PCL, many in vitro and in vivo studies for bone tissue reconstruction have been conducted. Various research results and references related to the PCL in vivo study in ‘Table 1’ should be described.

For example:

Macromol Biosci. 2021 Dec;21(12):e2100266. doi: 10.1002/mabi.202100266. Epub 2021 Oct 13.

Biomed Mater. 2020 Jul 1;15(4):045022. doi: 10.1088/1748-605X/ab843f.

Biomater Sci. 2021 Jun 7;9(11):4019-4039. doi: 10.1039/d1bm00062d. Epub 2021 Apr 26.

Acta Biomater. 2019 May;90:393-402. doi: 10.1016/j.actbio.2019.04.019. Epub 2019 Apr 6.

Acta Biomater. 2016 Jan;30:319-333. doi: 10.1016/j.actbio.2015.11.012. Epub 2015 Nov 10.

J Orthop Res. 2020 May;38(5):961-971. doi: 10.1002/jor.24542. Epub 2019 Dec 5.

Macromol Biosci. 2021 Mar;21(3):e2000336. doi: 10.1002/mabi.202000336. Epub 2020 Dec 21.

Materials (Basel). 2018 Feb 4;11(2):238. doi: 10.3390/ma11020238.

Int J Mol Sci. 2021 Mar 30;22(7):3588. doi: 10.3390/ijms22073588.

Gels 2022 Mar, 8:163. Doi:10.3390/gels8030163

Tissue Eng Part A. 2011 Jul;17(13-14):1831-9. doi: 10.1089/ten.TEA.2010.0560. Epub 2011 Apr 27.

Please search ‘pubmed.gov’ website.

Since there are many research cases on in addition and other materials (including chitosan, alginate, collagen, PLA), references and related information should be added.

 - Done, related information has been added

  1. In table 3, advantage section of bioactive glass and glass-ceramic was combined. Even if the content is the same, please write the core content separately for each bioceramic material.

 - Done

  1. There is also a selective laser melting technology that melts powder by using a higher-power laser, although it is similar to SLS (Selective laser sintering). At the end of ‘2.3.2. Selective laser sintering technology’ section, please also investigate and describe the SLM technology.

  - Done, related information has been added

  1. Fused deposition modeling (FDM), the most widely used 3D printing, that is a 3D printing method based on a continuous filament extrusion was not reported. Additional section of FDM is required.

  - Done, Additional section of FDM has been added

  1. Change the title of 2.3.3 to ‘Direct ink writing 3D printing technology’ in the same way as the contents.

- Done

Thanks
